# New Method for Imputation of Unquantifiable Values Using Bayesian Statistics for a Mixture of Censored or Truncated Distributions: Application to Trace Elements Measured in Blood of Olive Ridley Sea Turtles from Mexico

**DOI:** 10.3390/ani12212919

**Published:** 2022-10-25

**Authors:** Inmaculada Salvat-Leal, Adriana A. Cortés-Gómez, Diego Romero, Marc Girondot

**Affiliations:** 1Toxicology Area, Faculty of Veterinary Medicine, Regional Campus of International Excellence ‘Campus Mare Nostrum’, University of Murcia, Espinardo, 30100 Murcia, Spain; 2Laboratory Ecologie Systématique et Evolution, Université Paris-Saclay, CNRS, AgroParisTech, 91190 Gif-sur-Yvette, France

**Keywords:** detection limit, Bayesian model, censored distribution, truncated distribution

## Abstract

**Simple Summary:**

Analytical science in environmental research is frequently confronted with the problem of detection limits or missing data in the analyzed variables. This situation precludes the use of common methods of statistical analysis. We have developed a method to estimate the distribution of samples below or above the detection limit and were able to estimate the statistical distribution of the missing data. We test this method using a dataset of 25 trace elements measured in dead and alive marine turtles. We confirm previous finding that Cd and Na are significantly associated with dead or alive status, and we show that strontium concentration is also linked to this status.

**Abstract:**

One recurring difficulty in ecotoxicological studies is that a substantial portion of concentrations are below the limits of detection established by analytical laboratories. This results in censored distributions in which concentrations of some samples are only known to be below a threshold. The currently available methods have several limitations because they cannot be used with complex situations (e.g., different lower and upper limits in the same dataset, mixture of distributions, truncation and censoring in a single dataset). We propose a versatile method to fit the most diverse situations using conditional likelihood and Bayesian statistics. We test the method with a fictive dataset to ensure its correct description of a known situation. Then we apply the method to a dataset comprising 25 element concentrations analyzed in the blood of nesting marine turtles. We confirm previous findings using this dataset, and we also detect an unexpected new relationship between mortality and strontium concentration.

## 1. Introduction

Missing data are a common problem in the analytical sciences. In some fields, concentrations can be too low to be quantitatively detected because they are below the detection limit. For instance, this commonly occurs when determining the concentration of chemical elements in different samples. If the concentration is low, some laboratories cannot detect the presence of the chemical. However, it may still be present but at a concentration below the detection limit of the laboratory equipment. On the other hand, some concentrations may be too high, and detector saturation may prevent the accurate reporting of values [1]. The resulting data distributions are known as left and right cuts, respectively, and the limits of detection are the lower (LDL) and upper (UDL) detection limits. It is important to note that the LDL and UDL can simultaneously occur in a dataset. The measurements below the LDL or above the UDL are sometimes referred to “missing” [2], “nondetectable” [3], or “unobserved” [4]. However, “missing” data can be due to the detection limit or to other experimental problems that are unrelated to the limit of detection. Similarly, “nondetectable” or “unobserved” data can refer to the detection limit or to collected data that is incomplete or unable to be analyzed. Instead, the generic term used to refer to values below the LDL or above the UDL is “unquantifiable” data. This term will be used here and is abbreviated as UnQ.

Among these situations, three different cases are possible: the UnQ values may be reported as below or above the limits (for example in analytical sciences), simply not reported at all because the analysis does not permit to know that some samples are below or above the limits (for example in fisheries science when fishes are caught by nets with large meshes, small fishes are never caught), or reported as 0 (for example when electronic rectifiers are used). These three different situations are referred to as censored, truncated, and rectified distributions, respectively. However, the definitions vary, with the terms sometimes being used interchangeably [5].

Missing data are a key problem in statistical practice. Indeed, they are never welcome because most statistical methods cannot be applied directly to an incomplete dataset. The worst approach when dealing with UnQ is simply to exclude or delete them [6] as this produces a strong upward bias in all subsequent measures of location such as means and medians. Second-order statistics (standard error, standard deviation, range) are downward biased.

Two main strategies have been developed to handle datasets with UnQ data. In the first case, missing data are left as unquantified values, and only quantified ones are used. This can be achieved using conditional likelihood. Let x be a dataset and θ be a set of parameters. The likelihood of data x in a model with parameters θ is $L\left(x,\;\theta \right)$. This can then be easily converted into the likelihood of the data x being higher than a threshold LDL using $L\left(x>LDL,\;\theta \right)=L\left(x,\;\theta \right)/\int_{x=LDL}^{+\infty }L\left(x,\;\theta \right)dx$. This is the basis of maximum likelihood or Bayesian methods when managing unobserved states. Another category uses only comparisons between data with quantified values. The iconography of correlations can be used when UnQ data exist. The iconography of correlations was originally developed as a geometric method to search for links when multiple variables are studied in a single dataset [7,8]. Fundamentally, the coefficient of correlation R between two variables X and Y represents a geometric measurement for the dissimilarity of variations between these two variables. R is the cosine of the angle between vectors X and Y that is reduced and centered in the orthonormal space with N dimensions (N observations). This method can even be used in the presence of unobserved data because the correlations and partial correlations are made using all combinations of the three variables with quantified values [9]. On the other hand, it should be noted that UnQ are not unobserved data and imputation of UnQ can have still an interest in iconography of correlations.

Nevertheless, many statistical procedures require a complete dataset; for example, linear models, including general linear, general linearized, or general linear mixed models cannot be used with unobserved values. The same is true for multivariate analysis such as principal component analysis or its derivatives such as multiple discriminant analysis. One of the common approaches to dealing with missing values involves imputing missing values with plausible values. This leads to a complete dataset that can be analyzed by any statistical method. In environmental chemistry, the most common procedure to deal with UnQ data continues to be the substitution of some fraction of the detection limit. This method is better labeled as “fabrication” as it reports and uses a single value for concentration data where a single value is unknown [6]. LDL/2 is a commonly used substitution that assumes a uniform distribution below LDL [10]. If the data have a lognormal distribution, the substitution of $LDL/\sqrt{2}$ for each value below the LDL is preferred [10]. Some authors substitute UnQ data with 0 and with lower detection limit as the two extreme scenarios, and then calculate their statistics with both these substitution options under the idea that the truth should lie somewhere in between. This is clearly not a correct solution because the same dataset would then be used two times, which is considered invalid in statistics.

Nevertheless, these simple methods clearly have limits because they do not use all the data with observed values as a guide; only the detection limit is used. Furthermore, the methods cannot be used with UDL. A maximum likelihood estimation statistical method has been proposed based on the truncated normal distribution [10]. However, this method was described as “somewhat complex and requiring laborious calculations and use of tables” [10]. Recently, new multivariate methods were proposed to impute missing values in mixed datasets. One is based on the principal component method with the factorial analysis for mixed data, which balances the influence of all the variables that are continuous and categorical in the construction of the dimensions of variability. Because the imputation uses the principal axes and components, the prediction of the missing values is based on the similarity between individuals and the relationships between variables [11,12]. The method is available as an R package, missMDA, which handles missing values in multivariate data analysis [13]. An alternative is to use an iterative imputation method based on a random forest. By averaging values over many unpruned classification or regression trees, the random forest intrinsically constitutes a multiple imputation scheme. The method is also available as an R package, Missforest, a nonparametric missing value imputation for mixed-type data [14]. These multivariate methods are very powerful, but when missing data are produced by these methods, they cannot be used further in statistical tests because of the circularity of reasoning. When the multivariate procedure generates missing data that are used in a statistical test, it is not clear if the test result comes from the data or from the procedure used to generate them [15].

Considering this problem, we developed a method to replace the missing values. The specifications of this new method are as follows:To generate a value to replace UnQ data with a single value for all measurements or from a random value that respects the fitted distribution.To handle both the lower detection limit (LDL) and the upper detection limit (UDL).To deal with varying detection limits [16,17].To handle censored and truncated data.To fit a mixture model with several distribution models in a single dataset.

For the first four points, the method had to be able to manage heterogeneous datasets in a single analysis. Point 5 is important as the objective is to model the data with the best precision to model correctly the underlying distribution of observed values. A mixture model was implemented to consider that several processes could be at play at the same time.

Despite the ample literature for estimating distributions below or above a detection limit, surprisingly, many publications still use crude estimates such as LDL/2 or $LDL/\sqrt{2}$. Our objective is to provide an easy-to-use method to analyze samples with LDL or UDL but not at the expense of precision and robustness.

We used parametric modeling with five models: truncated normal distribution [10,18], lognormal distribution [4,18], Weibull distribution [19], gamma distribution [4], and generalized gamma distribution. Generalized gamma distribution has never been used in the ecotoxicological context until now. Each model is fitted using the maximum likelihood estimation with the conditional likelihood to estimate the LDL or UDL. If the data are censored, the proportion of quantified or UnQ data is also included in the model as a multinomial distribution. The different models can be compared using Akaike information criterion (AIC), which can be corrected for small samples (AICc) [20,21]. The fit of data from different origins can be compared using the Bayesian information criterion (BIC) [22]. The fitted maximum likelihood estimation parameters are used as a starting point for iterations using the Metropolis–Hastings algorithm with a Markov chain Monte Carlo (MCMC) in Bayesian analyses. The priors for the parameters are chosen from a wide uniform distribution. The Metropolis–Hastings algorithm is a MCMC method for obtaining a sequence of random samples from a probability distribution [23,24]. This method is now used widely as it offers a high-performance tool to fit a model. A goodness-of-fit test is implemented using the distribution of likelihoods of posterior predictive distribution compared with the likelihood of observed data in the fitted model. Finally, the model is scripted in R language and is available as an R package [25] as well as a simplified online tool (Available online: https://hebergement.universite-paris-saclay.fr/marcgirondot/cutter.html (accessed on 23 October 2022)) that is also available as a function iCutter() within the R package. The code can be checked within the source code of the package.

The method is tested with both fictive data and published data for metal concentrations analyzed in adult marine turtles (*Lepidochelys olivacea*) from Oaxaca state in Mexico [9].

## 2. Materials and Methods

### 2.1. Normal, Lognormal, Weibull, Gamma, and Generalized Gamma Distributions

The first step is to define the likelihood function to represent the probability that a sample has a concentration x. The general form is bell-shaped, with only positive continuous values and a right tail. Several functions have been used in the literature to represent this form, with truncated normal distribution [10,18], lognormal distribution [4,18], and gamma distribution [4] being the most common. We also included the Weibull and generalized gamma distributions.

The importance of normal distribution is partly related to the central limit theorem, which states that under some conditions, the average of many samples (observations) of a random variable with a finite mean and variance is itself a random variable whose distribution converges to a normal distribution as the number of samples increases. Therefore, physical quantities that are expected to be the sum of many independent processes often have nearly normal distributions. For a long time, the normal distribution was favored, because it was the only distribution with precise tables. However, with the current statistical languages and tools such as R or Python, this constraint has been removed, and a normal distribution is often not the best choice: its symmetrical nature and the truncation required so as to not generate negative concentrations make this choice irrelevant. A lognormal process is the statistical realization of the multiplicative product of many independent random variables, each of which is positive. The gamma distribution is the maximum entropy probability distribution (with respect to both a uniform base measure and a 1/x base measure) for a random variable X for which E[X] = kθ = α/β is fixed and greater than 0. The Weibull distribution is a continuous probability distribution [26] that is used in many scientific fields such as survival analysis, extreme value theory, or hydrology to measure extreme events. The generalized gamma distribution was introduced by Stacy [27] and included special sub-models such as the exponential, Weibull, gamma and Rayleigh distributions, among others. The generalized gamma distribution is appropriate for modeling data with dissimilar types of hazard rates [28]. It is important to note that these distributions are not used with the objective to model the process of contamination but rather to best represent the output of contamination.

The probability density of normal distribution is:$$f\left(x;\;\mu ,\;\sigma \right)=\frac{1}{\sigma \sqrt{2\pi }}e^{-\frac{1}{2}{\left(\frac{x-\mu }{\sigma }\right)}^{2}}$$
where μ is the mean and σ≥0 the standard deviation.

However, this form is not suitable to deal with concentrations because it can result in negative x. Therefore, a left truncated normal distribution must be used [29]:$$f\left(x\;|\;x>0;\;\mu ,\;\sigma \right)=e^{-\frac{1}{2}{\left(\frac{x-\mu }{\sigma }\right)}^{2}}/\int_{x=0}^{+\infty }e^{-\frac{1}{2}{\left(\frac{x-\mu }{\sigma }\right)}^{2}}\;dx$$

The probability density of lognormal distribution is:$$f\left(x;\;\mu ,\;\sigma \right)=\frac{1}{x\sigma \sqrt{2\pi }}e^{-\frac{{\left(log\left(x\right)-\mu \right)}^{2}}{2\sigma^{2}}}$$
where μ is the mean and σ≥0 the standard deviation.

The probability density of generalized gamma distribution can be written as follows [27]:$$f\left(x;\;a,\;d,\;p\right)=\frac{\left(p/a^{d}\right)x^{d-1}e^{-{\left(x/a\right)}^{p}}}{\Gamma \left(d/p\right)}$$
with x>0 and a, d, p>0 and where $\Gamma \left(k\right)$ is the gamma function of k. If d=p, then the generalized gamma distribution becomes the Weibull distribution [26]. Alternatively, if p=1, the generalized gamma becomes the gamma distribution.

The probability density functions can estimate the likelihood of one observation x in a distribution model by setting $L\left(x;\Theta \right)\;\#\;f\left(x;\Theta \right)$ where $f\left(x;\Theta \right)$ is one of the normal, lognormal, gamma, Weibull, or generalized gamma distributions and # indicates a proportional relationship. The parameters Θ are the generic representations of the required parameters depending on f.

### 2.2. LDL and UDL Conditioning Likelihoods

When these distributions are applied to concentration measurements, we must consider that only values between LDL and UDL have been really quantified. Then we can obtain the [LDL-UDL]-truncated distribution by:$$f\left(x\;|\;LDL<x<UDL;\Theta \right)=f\left(x;\Theta \right)/\int_{x=LDL}^{UDL}f\left(x;\Theta \right)\;\mathrm{dx}$$

This distribution can estimate the likelihood of one quantified observation x in a [LDL-UDL]-truncated distribution. Note that if LDL is 0, no truncation exists to the left and if UDL is +∞, no truncation exists to the right of the distribution.

This likelihood represents the likelihood of an observation in a truncated distribution. However, there is other information when a censored distribution is used. We know that $n_{LDL}$ samples are below the LDL and $n_{UDL}$ samples are above the UDL for a total of n samples. The expected proportion of $n_{LDL}$ among n in the $f\left(x;\Theta \right)$ distribution is thus:$$p_{LDL}=\int_{x=0}^{LDL}f\left(x;\Theta \right)\;dx$$

The expected proportion of $n_{UDL}$ among n in the $f\left(x;\Theta \right)$ distribution is thus:$$p_{UDL}=\int_{x=UDL}^{+\infty }f\left(x;\Theta \right)\;dx$$

Then the expected proportion of samples that have been successfully quantified is $n-n_{LDL}-n_{UDL}$, and their expected proportion is $1-p_{LDL}-p_{UDL}$.

The likelihood of the data $\left(n_{LDL},\;n_{LU}=n-n_{LDL}-n_{UDL},\;n_{UDL}\right)$ in the $\left(p_{LDL},\;p_{LU}=1-p_{LDL}-p_{UDL},\;p_{UDL}\right)$ model can be estimated using a multinomial distribution:$$f\left(n_{LDL},\;n_{LU},\;n_{UDL};n,\;p_{LDL},\;p_{LU},\;p_{UDL}\right)=\frac{n!}{n_{LDL}!n_{LU}!\;n_{UDL}!}\;p_{LDL}{}^{\;n_{LDL}}p_{LU}{}^{n_{LU}}p_{UDL}{}^{n_{UDL}}$$

If $n_{LDL}=0$ or $n_{UDL}=0$, the multinomial distribution becomes a binomial distribution.

In the case of a censored distribution, the likelihood of data in the model is thus the product of the likelihood of the quantified samples in the $f\left(x\;|\;LDL<x<UDL;\Theta \right)$ distribution with the likelihood of the proportion of UnQ samples based on a multinomial distribution.

### 2.3. Mixture Model

When data come from several sources, a mixture distribution model can be fitted using J sets of Θ parameters. Then we introduce a new set of weighting parameters named $q_{j}$ with j being the distribution model among the mixture with *J* distribution models with $\sum q_{j}=1$ with J−1 parameters. The j distribution has a contribution $q_{j}$ in the total distribution. If only one distribution is used, J=1 and $q_{1}=1$.

The likelihood of an observation $x_{i}$ is then:$$\overset{J}{\underset{j=1}{\sum }}q_{j}\;f\left(x_{i}\;|\;LDL<x<UDL;\Theta_{j}\right)$$

### 2.4. Maximum-Likelihood Estimation

The first step is to fit the parameters q and Θ that maximize the likelihood of data within the model $f\left(LDL<x<UDL;q,\;\Theta \right)$. The maximum-likelihood estimation is calculated using the quasi-Newton method [30], which allows box constraints. Quasi-Newton methods are a class of optimization methods used to find the global extremum of a function. The function *optim* from the *stats 4.2.1* R package was used (R Core Team and contributors worldwide).

When several models (normal, lognormal, gamma, Weibull, generalized gamma) are fitted to the same datasets of observations, the performance of the different models can be compared using AIC [20]. AIC is a measure of the quality of the fit that simultaneously penalizes for the number of parameters in the model. From a set of models, it facilitates selecting the best compromise between fit quality and over-parametrization:$$AIC_{j}=-2\ln L_{j}+2\;p_{j}$$
where *L_j_* is the likelihood of the model, and *p_j_* is the number of parameters of the model *j*.

When a set of *k* models are tested, the model with lowest AIC is the best non-overparametrized fit. It is important to note that the AIC value itself is not strictly informative in terms of the absolute model fit.

A corrected version of *AIC* for small sample sizes or second-order *AIC*, named *AICc*, has been proposed when the model is univariate and linear with normal residuals [31]:$$AICc=AIC+\frac{2\;p\;\left(p+1\right)}{n-p-1}$$

The formula can be difficult to determine when these conditions are not met, in which case, the previous formula can be used [21]. In general, the use of *AICc* is recommended instead of *AIC*, especially for datasets with small sample sizes.

When comparing a set of models, it is possible to estimate the relative probability that each model is the best among those tested using the Akaike weight [21]:$$Akaike\;weight_{j}=\frac{e^{\frac{1}{2}\;AIC_{j}-min\left(AIC\right)}}{\sum_{i=1}^{k}e^{\frac{1}{2}\;AIC_{i}-min\left(AIC\right)}}$$

We use *AIC*, *AICc*, and *Akaike weight* to compare the fitted distributions for our datasets.

The utility of model selection can be further extended to test for potential differences in the results from two or more datasets. In this case, the complete data are split into several subsets, with each individual dataset being represented once and only once. The test question is thus: can the collection of datasets be modeled with a single set of parameters, or must each dataset be modeled with its own set? In this situation, *BIC* should be used instead of *AIC* or *AICc* because the true model is obviously among the tested alternatives (if we consider that the distribution function plays a minor role compared with the fitted parameter value) [22]:BIC=−2lnL+plnn

When the BIC statistic is used, all the priors of the tested models are assumed to be identical. It is also possible to estimate *BIC* weights by replacing *AIC* with *BIC* in the Akaike weight formula. The *w*-value has been defined as the probability that several datasets can be correctly modeled by grouping them together instead of considering them independently [22].

### 2.5. Bayesian Fit Using the Metropolis-Hastings Algorithm

The Metropolis–Hastings algorithm is a MCMC method for obtaining a sequence of random samples from a probability distribution [23,24]. This method is now widely used, as it offers a high-performance tool to fit a model. To run Bayesian analysis with this algorithm, several parameters must be defined for each estimator in the model, particularly the priors. The priors represent the actual knowledge for the distribution of a parameter. It is beyond the scope of this paper to fully explore the fine details of this algorithm, and instead we focus on how to use it.

The choice of the prior is not straightforward [32] if only a few observations are available. A uniform distribution for priors indicates that all values within a range are equally probable, whereas a Gaussian distribution can use a mean and standard deviation obtained from previous analysis. Here we choose to use uniform distributed priors (see previous section).

During the iteration process, a Markov chain is constructed using the actual parameter values $\pi_{t}$ on which a new proposed random function defined by its standard deviation (*s*) is applied, $\pi_{t+1}=N\left(\pi_{t},\;s\right)$. This is the Monte Carlo process. The standard deviation (*s*) for a new proposal is a compromise between two constraints: if the values are too high, the new values could yield results far from the optimal solution, but if they are too low, the model can become stuck in local minima. The adaptive proposal distribution [33] ensures that the acceptance rate is around 0.234, which is close to the optimal rate in many realistic situations [34]. The burn-in value is the number of iterations necessary to stabilize the likelihood. Here, it was low (100) because the starting values are the maximum likelihood estimators. The total number of iterations was chosen to be 5000 after an initial diagnostic using 3000 iterations [35]. The Metropolis–Hastings algorithm implemented in HelpersMG packages available in CRAN [25] was used. The result of the MCMC analysis is a table with one set of values for the estimators at each iteration. The mean and standard deviation summary statistics can be calculated from this table using the coda R package [36], and it is also possible to estimate quantiles. The use of quantiles has the advantage of not requiring any hypothesis about the output distribution; hence, it is well-suited to an asymmetric distribution.

### 2.6. Posterior Predictive Distribution

The comparison of the likelihood of observed values within the fitted model and the likelihood of 5000 predictive posterior distribution generated with the MCMC result is used as a criterion of validity for the fitted model [37].

The posterior predictive distribution can be used also to generate a set of values with the UnQ replaced by values obtained from the posterior predictive distribution. It can be used to replicate an analysis and check if a particular conclusion is reached frequently or not (see below for its use using a generalized linear model).

### 2.7. Scripting the Model

The model was scripted in R language and included in HelpersMG packages available in CRAN [25]. The model can be run as the R function cutter() or as an online application using iCutter() or https://hebergement.universite-paris-saclay.fr/marcgirondot/cutter.html. One difficulty involved introducing a complex set of data that could be a mixture of censored or truncated data with various UnQ data either below the LDL or above the UDL. The usual value for missing data in R language is NA: “NA is a logical constant of length 1 which contains a missing value indicator” as defined in help (“NA”). However, UnQ is not missing information in the common sense; UnQ is rather information. Therefore, we introduce UnQ into the data as being −Inf for data below the LDL or +Inf for data above the UDL. When the dataset is homogeneous (same censored or truncated model and a single LDL and/or UDL for all the data), the data can be entered as a set of values and a single value for LDL in the lower-detection-limit parameter, UDL in the upper-detection-limit parameter, and cut model (censored or truncated) in the cut-method parameter. If the dataset is more complex, a data.frame object must be transmitted with four columns: Observations, LDL, UDL, and Cut (note that the column names can be changed).

Each observation is associated with LDL, UDL, and Cut information. It is also possible to define a distribution model to be fitted (normal, lognormal, gamma, Weibull, generalized gamma) as well as a number of distributions (1 to 9) when the mixture model is fitted. The cutter() function will return the posterior distribution for the missing LDL or UDL in this model to take the data into account. Associated functions are plot() to plot the results of a fitted model and rcutter() to generate random numbers from the fitted distribution.

When iCutter() or https://hebergement.universite-paris-saclay.fr/marcgirondot/cutter.html is used, the distribution model that best represents the data can be selected based on *AIC* and then fitted. However, this web-based model is a simplified version of the complete tools available in HelpersMG package because the models that include different LDL or UDL or a censored and truncated model in a single dataset cannot be fitted.

### 2.8. Sample Collection and Analysis

A fictive series of data was generated to test the methods using 100 values from a gamma distribution with k=10 and θ=20 and another set of 100 values with k=5 and θ=10 (using a 1234 seed for random numbers in R for reproducibility). In the first group, LDL was set to 20, and both truncated and censored series were simulated. In the second group, LDL was set to 20, and UDL was set to 300; in this case, only censored series were simulated because the truncated right series cannot be fitted. The results were also compared with Kaplan–Meier nonparametric estimations for the total distribution [38], as implemented in NADA v. 1.6–1.1 R package [39]. It should be noted that this method only allows for the global distribution of values rather than the estimation of missing data.

Another dataset comes from 238 blood samples collected from nesting females of the olive ridley sea turtle (*Lepidochelys olivacea*) at La Escobilla Beach in the state of Oaxaca (southeast Mexico, eastern Pacific, 96°44′ W and 15°47′ N). Beach monitoring was performed from August to September 2012 (nesting season, third arribada event), July to October 2013 (second to fifth arribadas), and August to October 2014 (fourth to sixth arribadas).

Metal concentrations were determined using an inductively coupled plasma optical emission spectrophotometer (ICP-OES, ICAP 6500 Duo Thermo^®^, Antigo, WI, USA). All concentrations are expressed in micrograms per gram in wet weight (or mg kg^−1^ ww). The LDL for all elements is 0.001 μg g^−1^. More details can be found in Cortés-Gómez, Romero, Santos, Rivera-Hernández and Girondot [9].

## 3. Results

### 3.1. Fictive Set of Data

The first group of left censored and truncated fictive data was fitted with normal, lognormal, Weibull, gamma, and generalized gamma with one or two distributions. The distribution has a mean of 130.06 and standard deviation of 90.62; these values do not differ from the values estimated using the Kaplan–Meier method that takes into account the standard error estimated using 1000 bootstraps: mean = 128.27 (SE = 6.29) and standard deviation = 90.23 (SE = 3.62). However, the Kaplan–Meier method cannot estimate missing values and is of no further interest here.

The Akaike weight and the goodness of fit for each tested model are shown in Table 1A (censored) and Table 1B (truncated). For the censored data, the two-mixture gamma distribution model is selected based on Akaike weight (0.32); the goodness of fit is 0.7024, indicating a 70.24% chance that the data could have been generated with this model. The second-order model by order of AIC is the Weibull distribution, with 0.24 Akaike weight and 0.5454 goodness of fit. The third-order model by order of AIC is the lognormal distribution, with 0.20 Akaike weight and 0.6956 goodness of fit. The plot of censored data and the selected model are shown in Figure 1A. When truncated data are fitted, a similar conclusion is obtained (Table 1B). The plot of censored data and the selected model are shown in Figure 1B. The posterior predictive mean of values below LDL are shown in Figure 2 for both the censored (A) and truncated (B) observations. The 95% credible intervals for lognormal, gamma, and generalized gamma distributions include the true values, but the 95% credible interval of the Weibull distribution does not (Figure 2).

Two advantages can be noted for the inclusion of LDL or UDL observations in the analysis. First, the 95% credible interval of the posterior predictive estimate is slightly smaller for the censored distribution compared with that of the truncated distribution. This can be observed in the bars around the median in Figure 2A versus Figure 2B. This allows more powerful tests to be performed. Second, the observed values could be a unimodal distribution, although the high proportion of LDL can be used to demonstrate that a bimodal distribution is hidden below the LDL. In an example with the same distribution but with LDL set to 100, only the right peak of data shown in Figure 1 is observed (Figure 3). However, when one or two gamma distributions are fitted and compared with AIC and Akaike weight, the model with two gamma distributions largely outperforms the model with only one distribution (one distribution: AIC = 1155.86, Akaike weight = 0.000; two distributions: AIC = 1129.56, Akaike weight = 1.000). If the same data are fitted without the LDL values (e.g., for a truncated distribution), a unimodal distribution will be selected (one distribution: AIC = 1113.86, Akaike weight = 0.840; two distributions: AIC = 1117.20, Akaike weight = 0.160).

Homogeneity between the two observed datasets can be tested by comparing likelihoods using BIC [22]. The data are derived from the gamma distribution with k=10 and θ=20 (distribution 1) and another set with k=5 and θ=10 (distribution 2). In group 1, 90% of the data come from distribution 1 and 10% from distribution 2, while in group 2, the reverse is true. The data were censored with LDL = 30. Group 1, group 2, and both groups together, named group 12, were fitted using the gamma distribution with a mixture of two distributions. Homogeneity for both datasets was tested using BIC weight. The fitted model with data from group 12 has a BIC of 2195.02, while data for group 1 and group 2 fitted separately have a BIC of 2077.98. The probability that a single model is sufficient to model these two groups is <0.0001, thus demonstrating the substantial difference between the two groups.

The second group of fictive data was left- (LDL = 20) and right-censored (UDL = 300) at the same time. These data were fitted with normal, lognormal, Weibull, gamma, and generalized gamma with one or two distributions. The selected model is a mixture of two gamma distributions with an Akaike weight of 0.40. The second and third models in order of Akaike weight are lognormal and normal distributions (0.28 and 0.27, respectively). The goodness of fit values for these three models are similar (>0.35). The plot of the selected model is shown in Figure 1B. The posterior predictive mean of values below the LDL and above the UDL are shown in Figure 4A,B. The 95% credible intervals for the selected lognormal, and gamma distributions include the true value, but the 95% credible interval of the normal truncated distribution does not.

It should be noted that when AIC or AICc was used, similar conclusions were reached.

### 3.2. Metal Concentration in Nesting Olive Ridleys

#### 3.2.1. Description of Database

An overview of the 25 analyzed elements in 238 samples of olive ridley sea turtles nesting in Oaxaca, Mexico, is shown in Table 2. A total of 12 elements had UnQ samples. Among these females, a total of 38 were found dead on the beach.

#### 3.2.2. Selected Distributions: Model and Mixture

The model that best describes the observations was selected for each of the 25 elements among censored gamma, lognormal, Weibull, and generalized gamma with one, two, or three distributions. Normal distribution was excluded, because it could produce biased estimations of UnQ values (see previous section). Lognormal was selected for 14 elements, gamma for 6, and generalized gamma for 5. The Weibull distribution was never selected (Table 3). For all elements, a mixture model with two distributions was selected, thus indicating that the distribution of the concentration of elements shares complex information.

#### 3.2.3. Homogeneity Test for Alive and Dead Turtles

A difference in concentration between alive and dead turtles was detected for 11 out of 25 elements (probability < 0.001 that the same distribution could be used to model these two categories) (Table 3).

#### 3.2.4. Generalized Linear Models According to Alive Status

The concentration of the 25 elements was tested to determine the relationship between dead and alive status. All the UnQ observations were replaced with a random number obtained from the posterior predictive distribution for each element. The posterior predictive distribution was the distribution fitted with all the observations, not the distribution fitted according to the alive or dead status of the turtle, so as not to bias the results. Then the generalized linear model with a binomial distribution and logit link was applied. Year (as a factor) was included as an interaction for each element. Model selection using AIC corrected from small samples (i.e., AICc) was used to simplify the model. This procedure was repeated 1000 times to generate the posterior predictive distribution of elements to show a relationship with alive status.

All the 1000 replicates gave the same selected elements and factor: Cd and Na (Figure 5). It is interesting to note that a relatively high frequency of false positives is observed when the data are shuffled, being around 20% (gray bars in Figure 5). Fitted relationships between alive and dead status and Cd and Na concentrations are shown in Figure 6. A higher concentration of Cd and a lower concentration of Na were observed in the blood of individuals found dead on the beach in both 2012 and 2013.

## 4. Discussion

The objective of the present work was to develop a method to estimate the distribution of samples below or above the detection limit, named here “unquantifiable” data and abbreviated as UnQ. The model used both conditional maximum likelihood and Bayesian statistics. Using fictive data of known distribution, we showed that the distribution model and the estimated UnQ value were correct. Two conclusions arise from this analysis: similar conclusions were reached with the AIC or AICc model selection, whereas the truncated normal distribution could produce some bias in the UnQ estimation. For this reason, the truncated normal distribution was not used in the subsequent analyses.

A dataset containing the concentrations of 25 elements measured in the blood of female marine turtles nesting in Oaxaca (Mexico) was analyzed. A total of 238 nesting females were sampled, 200 apparently healthy nesting turtles and 38 recently dead individuals (half of them from unknown causes and the other half due to boat collisions and bycatch). The selected distribution models were lognormal, gamma, or generalized gamma (Table 3); the truncated normal distribution was not tested (see previous section). The Weibull model was never selected, although there is no theoretical reason not to test this distribution. It is important to recall that the choice of distributions aims only to reproduce a pattern but not to explain the origin of this pattern.

For all the elements, a mixture of two distributions was selected rather than a single distribution (Table 3). For some elements, a mixture of three distributions had a lower *AICc* than a mixture of two distributions, although the amelioration of the model was very low when measured by Akaike weight. For this reason, only mixtures of two distributions were further analyzed.

A homogeneity test was performed for each element to detect if a single model is sufficient to model the two categories of females (dead and alive) or if two different models should be used. Eleven elements were identified (Ca, Cd, Fe, K, Mg, Na, P, Se, S, Sr, and Zn). Note that this test could detect elements without a direct link to alive status, although the link could be mediated by other elements or factors.

Then a generalized linear model was set up using all the elements in a single analysis as well as the interaction with sampling year as a factor. The model was then simplified using *AICc*. Overall, 1000 replicates were performed using UnQ values that were replaced by a random number obtained from posterior predictive distribution. In all the 1000 replicates, only two elements were selected: Cd and Na. The other previously identified elements were never selected (Ca, Fe, K, Mg, P, Se, S, Sr, and Zn). Cd concentration was higher in turtles that died on the beach, while Na concentration was lower in these individuals (Figure 6). These findings are concordant with the results found using a completely different method, namely the iconography of correlations [9]. The two elements Cd and Na identified as being significantly associated with dead or alive status were also identified as being in direct interaction with this status using the iconography of correlations method [9]. Among the 11 elements that were better modeled with two distributions, 10 were shown to be linked with Na concentration using the iconography of correlations and then indirectly with alive or dead status. The very similar results obtained by two innovative and unrelated methods can be used to validate both. The only element that we detected to be better modeled using two distributions for dead and alive individuals, which was not found in a previous study, is Strontium (Sr). Sr is an inorganic non-essential element that occurs naturally in igneous rocks. Its concentration was higher in dead compared with alive turtles (dead: mean = 1.82, median = 1.11, max = 9.23; alive: mean = 1.06, median = 0.99, max = 3.13; Table 2 in [9]). Sr is chemically similar to Ca and is often incorporated into bone and eggshells by organisms [40]. Typically, Ca and Sr incorporated into the egg originates from the bones and blood of the female [41]. High concentrations of Sr in avian eggshells have been associated with reduced hatching success [42] and increased egg breakage and possibly bone deformities [43]. We do not have information about the toxicity of Sr in marine turtles [44], although more attention should clearly be given to this element in the future.

## 5. Conclusions

Analytical science in environmental research is frequently confronted with the problem of detection limits or missing data in the analyzed variables. This situation precludes the use of common methods of statistical analysis. The objective of the present work was to develop a method to fill in these gaps. Using both conditional maximum likelihood and Bayesian statistics, we were able to build a model to respond to our objectives: (1) to generate a value to replace UnQ data with a single value for all measurements or a random value respecting the fitted distribution; (2) to handle both the lower detection limit (LDL) and the upper detection limit (UDL); (3) to deal with varying detection limits; (4) to handle censored and truncated data; and (5) to fit a mixture model with several distribution models in a single dataset. However, in our opinion, the main difficulty was being able to provide a method that can be used by researchers without a strong background in statistics. This was achieved by developing an online application in which the user provides the series of values and the limit of detection, with all the tests being performed automatically.

## Figures and Tables

**Figure 1 animals-12-02919-f001:**
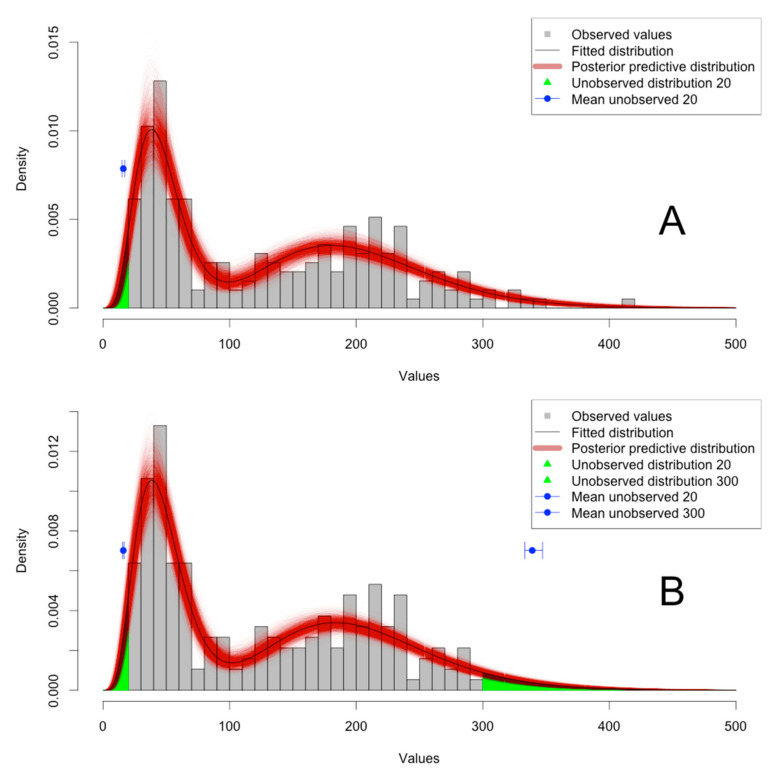
Observed distribution of the fictive set of censored left (LDL = 20; (**A**)) and left and right (LDL = 20, UDL = 300; (**B**)) data with the corresponding Akaike information criterion AIC-selected model for gamma distribution. Posterior predictive distribution based on 5000 Markov chain Monte Carlo iterations is shown along with the mean and 95% credible interval for values below the lower detection limit (LDL) (**A**,**B**) and above the upper detection limit (UDL) (**B**).

**Figure 2 animals-12-02919-f002:**
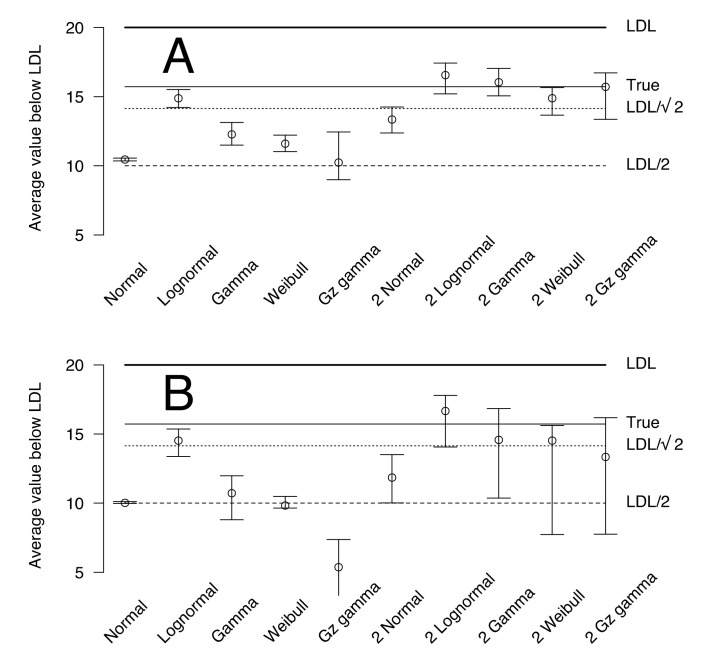
Average and 95% credible interval of the fitted values below the lower detection limit (LDL) for the censored (**A**) and truncated (**B**) fictive distributions. The models are shown on the *x-axis*, and the corresponding fitted values are shown on the *y-axis*. The true value based on the distribution used to generate the data is shown on the right, along with the usual LDL/2 and $LDL/\sqrt{2}$ values used for the substitution.

**Figure 3 animals-12-02919-f003:**
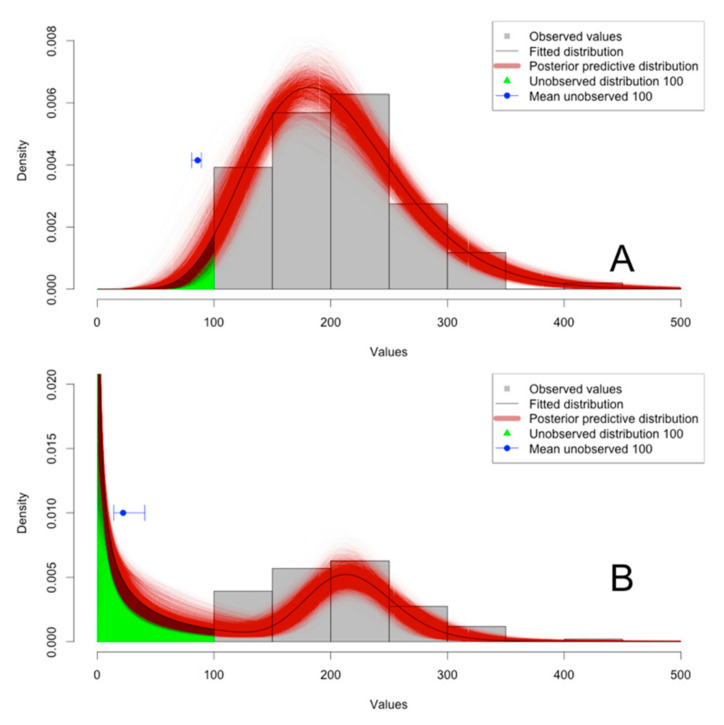
Observed distribution of the fictive set of censored left (LDL = 100) data with the corresponding fitted model for one (**A**) or two (**B**) gamma distributions. The model uses truncated data in A and censored data in B to demonstrate the information shared by the lower detection limit (LDL) observations. Posterior predictive distribution based on 5000 Markov chain Monte Carlo iterations are shown along with the mean and 95% credible interval for the values below the LDL.

**Figure 4 animals-12-02919-f004:**
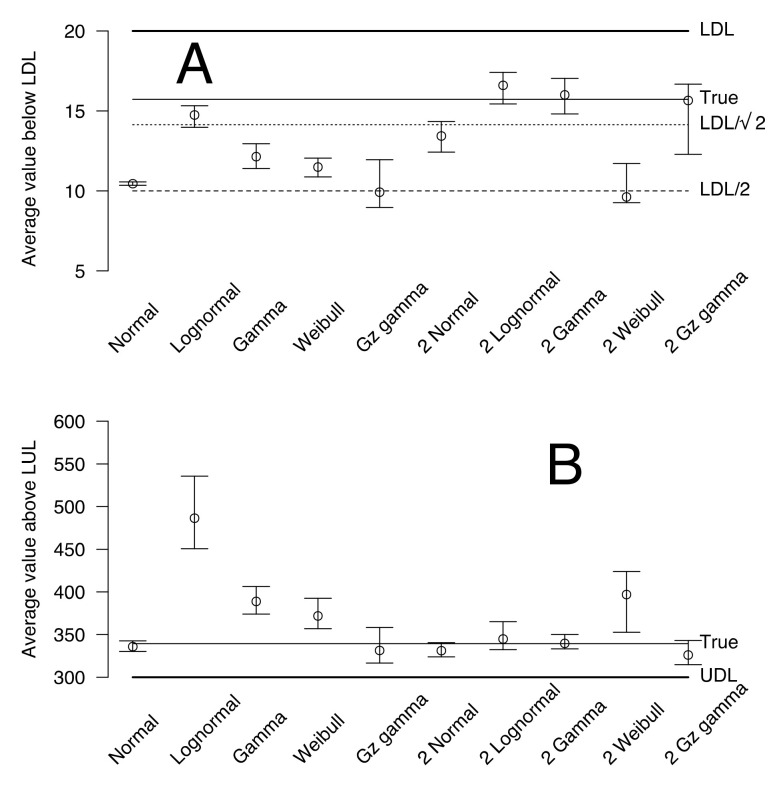
Average and 95% credible interval of the fitted values below the lower detection limit (LDL) (**A**) and above the upper detection limit (UDL) (**B**) fictive distributions. The models are shown on the *x*-axis, and the corresponding fitted values are shown on the *y*-axis. The true value based on the distribution used to generate the data is shown on the right, along with the usual LDL/2 and $LDL/\sqrt{2}$ values for LDL substitution.

**Figure 5 animals-12-02919-f005:**
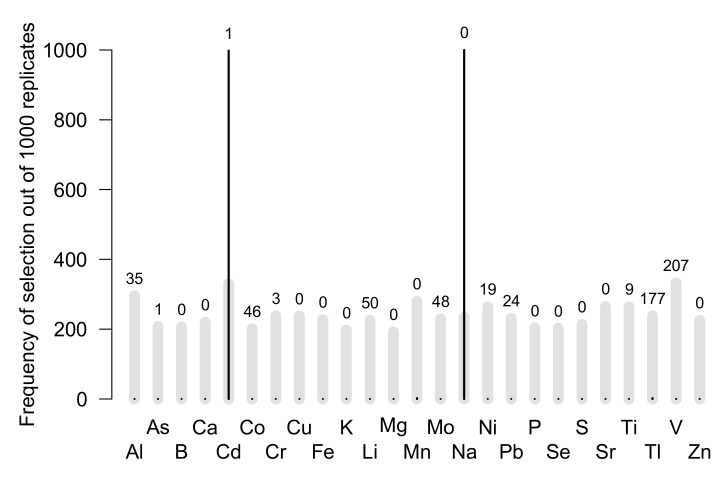
Frequency of selection for each of the 25 analyzed elements (in abscise) out of 1000 replicates in a generalized linear model linking alive status with the concentration of the elements. Unquantifiable (UnQ) values are replaced with a posterior predictive random value in replicates. Model selection was performed using AICc (i.e., Akaike information criterion with a small sample correction). Black lines correspond to the number of times the element is selected, and gray lines are the same when data are shuffled (false positive). The number at the top of each bar is the number of UnQ.

**Figure 6 animals-12-02919-f006:**
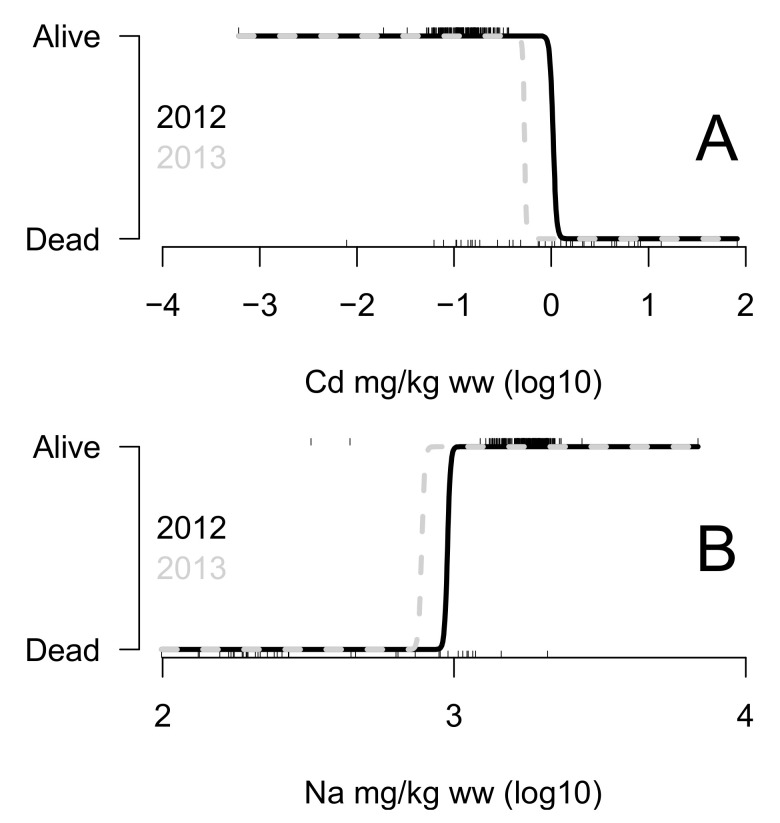
Relationship between alive and dead status and concentration of Cd (**A**) and Na (**B**) in *Lepidochelys olivacea* female blood (mg/kg wet weight). Dashes are observations, and lines are the fitted models using the generalized linear model with year of sampling as the interaction.

**Table 1 animals-12-02919-t001:** Model selection based on Akaike information criterion (AIC), Akaike weight, and goodness of fit (GoF) for normal, lognormal, gamma, Weibull, and generalized gamma distributions applied to a set of 200 left-censored (A) or left-truncated (B) fictive data. Selected models based on a combination of Akaike weight and GoF are highlighted in bold.

A: Censored					
Distribution	Mixture	AIC	∆AIC	Akaike Weight	GoF
Normal	1	2288.58	110.06	0.00	0.0000
Lognormal	1	2252.10	73.57	0.00	0.3700
Gamma	1	2241.80	63.28	0.00	0.1136
Weibull	1	2240.31	61.78	0.00	0.0674
Generalized gamma	1	2239.63	61.10	0.00	0.0530
Normal	2	2179.53	1.00	0.19	0.3970
**Lognormal**	**2**	**2179.47**	**0.95**	**0.20**	**0.6956**
**Gamma**	**2**	**2178.52**	**0.00**	**0.32**	**0.7024**
**Weibull**	**2**	**2179.08**	**0.56**	**0.24**	**0.5454**
Generalized gamma	2	2182.15	3.48	0.06	0.6516
**B: Truncated**					
**Distribution**	**Mixture**	**AIC**	**∆AIC**	**Akaike Weight**	**GoF**
Normal	1	2220.66	45.61	0.00	0.5402
Lognormal	1	2247.81	72.77	0.00	0.5236
Gamma	1	2229.30	54.26	0.00	0.5426
Weibull	1	2226.84	51.80	0.00	0.5186
Generalized gamma	1	2203.15	28.11	0.00	0.5586
**Normal**	**2**	**2208.00**	**0.02**	**0.26**	**0.8746**
**Lognormal**	**2**	**2175.64**	**0.60**	**0.20**	**0.7292**
**Gamma**	**2**	**2175.04**	**0.00**	**0.27**	**0.7458**
**Weibull**	**2**	**2175.42**	**0.38**	**0.22**	**0.7576**
Generalized gamma	2	2178.36	3.34	0.05	0.8018

**Table 2 animals-12-02919-t002:** Number of unquantifiable (UnQ) samples for the 25 analyzed elements in 238 blood samples of olive ridley sea turtles nesting in Oaxaca, Mexico [9].

Element	Number of UnQ	Proportion of UnQ	Element	Number of UnQ	Proportion of UnQ
Al	35	0.147	Mo	48	0.202
As	1	0.004	Na	0	0.000
B	0	0.000	Ni	19	0.080
Ca	0	0.000	Pb	24	0.101
Cd	1	0.004	P	0	0.000
Co	46	0.193	Se	0	0.000
Cr	3	0.013	S	0	0.000
Cu	0	0.000	Sr	0	0.000
Fe	0	0.000	Ti	9	0.038
K	0	0.000	Tl	177	0.744
Li	50	0.210	V	207	0.870
Mg	0	0.000	Zn	0	0.000
Mn	0	0.000			

**Table 3 animals-12-02919-t003:** Comparison of fit of censored gamma, lognormal, Weibull, and generalized gamma distributions for the elements analyzed in 238 blood samples of olive ridley sea turtles nesting in Oaxaca, Mexico [9]. Only the selected model based on Akaike weight using AICc (i.e., Akaike information criterion with a small sample correction) is shown. The column “Test dead vs. alive” shows the Akaike weight based on the Bayesian information criterion (BIC) for a single model for both alive and dead turtles on the beach.

	Distribution Model Selection			Test Dead vs. Alive
Element	Selected Distribution	Mixture	Akaike Weight Based on AICc	w-Value (BIC) for Single Model
Al	lognormal	2	1.00	1.000
As	generalized gamma	2	0.74	1.000
B	lognormal	2	0.60	1.000
Ca	lognormal	2	0.98	0.000
Cd	lognormal	2	1.00	0.000
Co	gamma	2	0.40	1.000
Cr	gamma	2	0.44	0.250
Cu	lognormal	2	0.58	0.120
Fe	gamma	2	0.32	0.000
K	lognormal	2	1.00	0.000
Li	generalized gamma	2	0.99	1.000
Mg	lognormal	2	0.94	0.000
Mn	lognormal	2	0.99	0.999
Mo	generalized gamma	2	0.92	1.000
Na	lognormal	2	0.71	0.000
Ni	generalized gamma	2	0.72	1.000
Pb	lognormal	2	0.56	1.000
P	lognormal	2	1.00	0.000
Se	gamma	2	0.31	0.000
S	lognormal	2	0.99	0.000
Sr	lognormal	2	0.64	0.000
Ti	lognormal	2	1.00	1.000
Tl	gamma	2	0.50	1.000
V	gamma	2	0.46	1.000
Zn	generalized gamma	2	0.78	0.000

## Data Availability

The complete dataset is available upon request to the corresponding author.

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
