# Peer review of "New Method for Imputation of Unquantifiable Values Using Bayesian Statistics for a Mixture of Censored or Truncated Distributions: Application to Trace Elements Measured in Blood of Olive Ridley Sea Turtles from Mexico"

_animals, 2022, doi:10.3390/ani12212919_

Round 1
Reviewer 1 Report
Thank you for the opportunity to review this manuscript which touches upon a very relevant and important topic.
I only have one major hesitation with the presented approach. In L. 136 it is mentioned that “the fitted max likelihood parameters are then used as prior for Bayesian analysis”. Do I understand correctly that the authors propose to first fit the 5 different models (and mixtures of these) using frequentist maximum likelihood inference (quasi-Newton optimization), and then use the resulting parameter estimates as priors in a Bayesian inference process (which then aims to again obtain parameter estimates on the same set of data). This is problematic, because, the same dataset would then be used 2 times, first for frequentist max likelihood parameter estimation, and then again for Bayesian parameter estimation. This repeated use of the same data to inform the prior as well as the “independent” dataset in Bayesian inference is a serious problem. If one uses part of a dataset to inform the priors, these datapoints should be deleted before the rest of the dataset can be used as independent data in the Bayesian inference process. I hope this is a misunderstanding from my part. Please explain in more detail how data are used between the maximum likelihood parameter inference and the Bayesian parameter inference.
I have some additional minor comments to improve readability and clarity of the manuscript, which overall reads nicely and presents the problem, approaches and methods clearly:
L. 36: “safely” does not seem an appropriate word. Perhaps “accurately” or “precisely” or “quantitatively”.
L. 37: please add: “concentration of” in ...determining the concentration of chemical elements...
L. 41: The authors use “distributions” here and in some other places throughout the manuscript (e.g. L. 54). However, a distribution somehow presents a representation of the data, and already makes an assumption that these data come from a distribution. I think that in this place “dataset” is more appropriate, or alternatively “The resulting data distributions are respectively known as..” (in L. 54: censored, truncated, and rectified datasets. In L. 120 you refer to them as censored and truncated data).
L. 44 – 62: Thank you for this section. It is very good to clear up the terminology (and indicate why it might be wrong), and the problematics of how people report UnQ.
L. 45. “labelled” might be misleading because it sounds as if you would put a value in the dataset. Perhaps “referred to” is more suitable.
L. 53. “set to 0” might better be “reported as 0”, because later (L90) you explain how people “fabricate” data by setting them to LOD/2 etc... (When you wrote “set to 0”, I expected immediately to also read about these other options). Alternatively, “set to an arbitrary value”
L 53: Just to be very clear how you distinguish between censored and truncated (which you also refer to in L 120), do you mean with “simply not reported” that the detection limits are not given at all (also not in e.g. an appendix or separate document)? Perhaps it is worthwhile making this explicit: “... reported as below or above the limits (with the value of the detection limit given), simply not reported at all (where the detection limits are not given), or reported as 0 (with the detection limit not given???).
L 63-80: this is interesting, and thanks for showing these different approaches. However, in the logic and flow of the introduction, wouldn’t it better fit together with section L 93-112 where the authors explain different approaches (maximum likelihood, multivariate methods, random forests)?
L. 92: Perhaps it is worthwhile mentioning that people also often substitute with 0 and with LOD as the two extreme scenarios, and then calculate their statistics with both these substitution options under the idea that the truth should lie somewhere in between. Also, people sometimes combine several approaches (e.g. as the authors discuss L 545) as a work around the problem of censored data.
L. 121: from the introduction, up until this point, it is clear to me why you want to address points 1 to 4. It is less clear why you need point 5. What is the advantage of a mixture model? Please add a few sentences. (also, L. 134 mentions that the different models are compared, so it is unclear to me from this text whether you fit 1 mixture model, or simply several models). ...after reading the rest of the paper, it becomes clearer, but it is still useful to be clear at this point in the manuscript.
L 230-234: This section is a bit difficult to follow. It is not clear what the parameters q are (is this a weighting parameter for each model?). “data come from different sources”-> as far as I see, you only use 1 dataset, so perhaps this needs a different wording. (very minor comment: Why not use capital J as the vector of distribution models, where small j is one of the models, i.e. use J instead of k to not have to introduce another letter; )
L. 240. Perhaps a few words on how the quasi-Newton method works (and which one specifically you applied) would be valuable. Alternatively, did you use an R function or package for this, it might be nice to mention this.
L290 – 295: I do not think it is necessary to explain how MCMC works, I would rather refer to any Bayesian inference textbook for that (L 291 parameters π come out of nowhere).
L. 278: what r package was used for the inference. (You mention the coda package for postprocessing)
L. 489: This replacement procedure is an important contribution of this paper and approach, because this is what sets it apart from e.g. Kaplan Meier which can only provide summary statistics. Hence, it is worthwhile to explain this a bit more clearly (and perhaps also explain it much earlier in the methods section, rather than having it get lost at the end of the results section). Do you mean that for each UnQ observation you take a sample from the part of the posterior probability distribution that is below detection limit, with the probability of a given concentration for a given UnQ observation depending on the probability distribution in this censored part of the posterior distribution.
Author Response
Reviewer 1
Thank you for the opportunity to review this manuscript which touches upon a very relevant and important topic.
I only have one major hesitation with the presented approach. In L. 136 it is mentioned that “the fitted max likelihood parameters are then used as prior for Bayesian analysis”. Do I understand correctly that the authors propose to first fit the 5 different models (and mixtures of these) using frequentist maximum likelihood inference (quasi-Newton optimization), and then use the resulting parameter estimates as priors in a Bayesian inference process (which then aims to again obtain parameter estimates on the same set of data). This is problematic, because, the same dataset would then be used 2 times, first for frequentist max likelihood parameter estimation, and then again for Bayesian parameter estimation. This repeated use of the same data to inform the prior as well as the “independent” dataset in Bayesian inference is a serious problem. If one uses part of a dataset to inform the priors, these datapoints should be deleted before the rest of the dataset can be used as independent data in the Bayesian inference process. I hope this is a misunderstanding from my part. Please explain in more detail how data are used between the maximum likelihood parameter inference and the Bayesian parameter inference.
We agree with the remarks of the referee about the remark that the same data should not be used 2 times.
The sentence line 136 “The fitted maximum likelihood estimation parameters are used as a prior for Bayesian analysis” was misleading. Indeed, the point estimation obtained from ML was used as mean value for normal prior distribution, but the standard deviation was chosen to be very large to ensure that normal distribution didn’t constrain too much the posterior distribution. The default has been changed to use uniform priors and the results of the first analysis using frequentist maximum likelihood inference (quasi-Newton optimization) is used simply as a starting point for the MCMC iterations. It permits to reduce the need of long burn-in that is time consuming (0 iteration was used but it has been changed to 100 iterations to take into account that ML is not a random starting point). The manuscript is changed accordingly, and all the analyses are done again with this new setup.
The fitted maximum likelihood estimation parameters are used as a starting point for iterations using a Metropolis–Hastings algorithm with a Markov chain Monte Carlo (MCMC) in Bayesian analyses. The priors for the parameters are chosen from wide uniform distributions.
I have some additional minor comments to improve readability and clarity of the manuscript, which overall reads nicely and presents the problem, approaches and methods clearly:
- 36: “safely” does not seem an appropriate word. Perhaps “accurately” or “precisely” or “quantitatively”.
The word safely is changed to quantitatively.
- 37: please add: “concentration of” in ...determining the concentration of chemical elements...
Change has been done.
- 41: The authors use “distributions” here and in some other places throughout the manuscript (e.g. L. 54). However, a distribution somehow presents a representation of the data, and already makes an assumption that these data come from a distribution. I think that in this place “dataset” is more appropriate, or alternatively “The resulting data distributions are respectively known as..” (in L. 54: censored, truncated, and rectified datasets. In L. 120 you refer to them as censored and truncated data).
We agree that the term distribution is not pertinent for the data themselves. We change the sentence to:
The resulting data distributions are respectively known as left and right cuts, while the limits of detection are the lower (LDL) and upper (UDL) detection limits.
- 44 – 62: Thank you for this section. It is very good to clear up the terminology (and indicate why it might be wrong), and the problematics of how people report UnQ.
Thanks
- 45. “labelled” might be misleading because it sounds as if you would put a value in the dataset. Perhaps “referred to” is more suitable.
Change has been done.
- 53. “set to 0” might better be “reported as 0”, because later (L90) you explain how people “fabricate” data by setting them to LOD/2 etc... (When you wrote “set to 0”, I expected immediately to also read about these other options). Alternatively, “set to an arbitrary value”
At this point of the text, we are still describing how authors report data. The use of LOD/2 comes in a second step during the analysis.
L 53: Just to be very clear how you distinguish between censored and truncated (which you also refer to in L 120), do you mean with “simply not reported” that the detection limits are not given at all (also not in e.g. an appendix or separate document)? Perhaps it is worthwhile making this explicit: “... reported as below or above the limits (with the value of the detection limit given), simply not reported at all (where the detection limits are not given), or reported as 0 (with the detection limit not given???).
We have tried to be more precise to identify censored and truncated distribution:
Among these situations, three different cases are possible: the UnQ values may be reported as below or above the limits (for example in analytical sciences), simply not reported at all because the analysis does not permit to know that some samples are below or above the limits (for example in fisheries science when fishes are caught by nets with large meshes; small fishes are never caught), or reported as 0 (for example when electronic rectifier are used).
L 63-80: this is interesting, and thanks for showing these different approaches. However, in the logic and flow of the introduction, wouldn’t it better fit together with section L 93-112 where the authors explain different approaches (maximum likelihood, multivariate methods, random forests)?
At this point of the text, we are still describing the structure of data. Line 93-112 explain what is generally done for the analysis of these data. So we prefer to maintain this structure.
- 92: Perhaps it is worthwhile mentioning that people also often substitute with 0 and with LOD as the two extreme scenarios, and then calculate their statistics with both these substitution options under the idea that the truth should lie somewhere in between. Also, people sometimes combine several approaches (e.g. as the authors discuss L 545) as a work around the problem of censored data.
We have added this precision.
Some authors substitute UnQ data with 0 and with lower detection limit as the two extreme scenarios, and then calculate their statistics with both these substitution options under the idea that the truth should lie somewhere in between. This is clearly not a correct solution because the same dataset would then be used 2 times which should be banned in statistics.
- 121: from the introduction, up until this point, it is clear to me why you want to address points 1 to 4. It is less clear why you need point 5. What is the advantage of a mixture model? Please add a few sentences. (also, L. 134 mentions that the different models are compared, so it is unclear to me from this text whether you fit 1 mixture model, or simply several models). ...after reading the rest of the paper, it becomes clearer, but it is still useful to be clear at this point in the manuscript.
Right. The objective was to model the data with the best precision to model correctly the underlying distribution of observed values. A mixture model was implemented to test the hypothesis that several processes were at play to determine the observed data. This precision is added:
The point 5 is important as the objective is to model the data with the best precision to model correctly the underlying distribution of observed values. A mixture model was implemented to consider that several processes could be at play at the same time.
L 230-234: This section is a bit difficult to follow. It is not clear what the parameters q are (is this a weighting parameter for each model?).
Right. This precision is added:
Then we introduce a new set of weighting parameters…
“data come from different sources”-> as far as I see, you only use 1 dataset, so perhaps this needs a different wording. (very minor comment: Why not use capital J as the vector of distribution models, where small j is one of the models, i.e. use J instead of k to not have to introduce another letter; )
The referee is right that k can be renamed J to not introduce a new letter. Change has been done.
When data come from several sources, a mixture distribution model can be fitted using sets of parameters. Then we introduce a new set of weighting parameters named with being the distribution model among the mixture with J distribution models with with parameters. The distribution has a contribution in the total distribution. If only one distribution is used, and .
The likelihood of an observation is then:
- 240. Perhaps a few words on how the quasi-Newton method works (and which one specifically you applied) would be valuable. Alternatively, did you use an R function or package for this, it might be nice to mention this.
The precision is added:
The maximum-likelihood estimation was made using the quasi-Newton method [30], which allows box constraints. Quasi-Newton methods are a class of optimization methods used to find the global extremum of a function. The function optim from the stats 4.2.1 R package was used.
L290 – 295: I do not think it is necessary to explain how MCMC works, I would rather refer to any Bayesian inference textbook for that (L 291 parameters π come out of nowhere).
The description of MCMC is very short in the manuscript and we don’t think that it can be easily shorten as it is necessary to indicate the conditions of use (burn-in and number of iterations). No change has been done.
- 278: what r package was used for the inference. (You mention the coda package for postprocessing)
The precision is added:
The Metropolis-Hastings algorithm implemented in HelpersMG packages available in CRAN [25] was used.
- 489: This replacement procedure is an important contribution of this paper and approach, because this is what sets it apart from e.g. Kaplan Meier which can only provide summary statistics. Hence, it is worthwhile to explain this a bit more clearly (and perhaps also explain it much earlier in the methods section, rather than having it get lost at the end of the results section). Do you mean that for each UnQ observation you take a sample from the part of the posterior probability distribution that is below detection limit, with the probability of a given concentration for a given UnQ observation depending on the probability distribution in this censored part of the posterior distribution.
We add this precision in the section 2.5 of the material and method:
2.5. Posterior Predictive Distribution
The comparison of the likelihood of observed values within the fitted model and the likelihood of 5,000 predictive posterior distribution generated with the MCMC result is used as a criterion of validity for the fitted model [37].
The posterior predictive distribution can be used also to generate set of values with the UnQ replaced by values obtained from the posterior predictive distribution. It can be used to replicate an analysis and check if a particular conclusion is reached frequently or not (see below for its use using generalized linear model).

Reviewer 2 Report
This paper focuses on the issue of missing data in analytical work where methods of detection themselves have limits. The authors have created an ap or stats package that makes up data where none is available from limitations of assays used. They have used a relatively crude separation of alive vs dead animals and shown some elements seem associated with the state of the animal at time of sampling, particularly with regard to Cd and Na and maybe strontium.
This reviewer has concerns with the value of the findings but see's no reason the authors should not be allowed to present them.
specific comments:
Line 114 method, not methodology which is study of method
Section 2 provides a basic statistical review. Depends on audience expected to a large degree how useful or necessary this is. Cursory read did not identify any errors in the proffered equations, but reviewer did not check back to all cited sources.
In 2.5 the choice to use normally distributed priors is interesting and potentially rather controversial. The author’s perception of normal distributions being common in wildlife situations is counter to the reviewer’s experience where such findings would best be described as extremely rare. The authors have a right to their opinion but it is not necessarily factual.
Line 296 The statement that 0.234 is the optimal rate is stark and interesting but hardly biblical.
Author Response
Reviewer 2
This paper focuses on the issue of missing data in analytical work where methods of detection themselves have limits. The authors have created an ap or stats package that makes up data where none is available from limitations of assays used. They have used a relatively crude separation of alive vs dead animals and shown some elements seem associated with the state of the animal at time of sampling, particularly with regard to Cd and Na and maybe strontium.
We agree that the distinction between alive and dead animals is crude, but we still think that it is pertinent for a species conservation point of view.
This reviewer has concerns with the value of the findings but see's no reason the authors should not be allowed to present them.
specific comments:
Line 114 method, not methodology which is study of method
Change has been done
Section 2 provides a basic statistical review. Depends on audience expected to a large degree how useful or necessary this is. Cursory read did not identify any errors in the proffered equations, but reviewer did not check back to all cited sources.
In 2.5 the choice to use normally distributed priors is interesting and potentially rather controversial. The author’s perception of normal distributions being common in wildlife situations is counter to the reviewer’s experience where such findings would best be described as extremely rare. The authors have a right to their opinion but it is not necessarily factual.
I completely agree with the referee view (and I teach indeed that normal distributions are not pertinent in most situations!). As a default, in the current version I use uniform distribution to not constrain the posterior distribution. See answer to referee 1; text has been changed to:
The fitted maximum likelihood estimation parameters are used as a starting point for iterations using a Metropolis–Hastings algorithm with a Markov chain Monte Carlo (MCMC) in Bayesian analyses. The priors for the parameters are chosen from wide uniform distributions.
Line 296 The statement that 0.234 is the optimal rate is stark and interesting but hardly biblical.
This 0.234 comes from analysis done by Roberts et al. (1997).
Roberts, G.O., Gelman, A. & Gilks, W.R. (1997) Weak convergence and optimal scaling of random walk Metropolis algorithms. The Annals of Applied Probability, 7, 110-120.
To be more precise, the exact conclusion of this analysis can be written as:
If you have a d-dimensional target that is independent in each coordinate, then choose the step size of random walk kernel to be 2.38 / sqrt(d).
However, the targets are never independent in each coordinate, and it was then proposed to tune the acceptance rate to be around 1/4.
We change the sentence to be less biblical:
The adaptive proposal distribution [33] ensures that the acceptance rate is around 0.234, which is close to the optimal rate in many realistic situations [34].

Round 2
Reviewer 1 Report
All my comments and concerns were adressed properly.
I can see now the distinction between using the MLE parameters to inform on the starting values, and using uniform prior parameter distributions in the Bayesian inference. This clears up my main concern about the manuscript.
Note: L 299 still mentions: "Here we choose to use normally distributed priors inferred from the maximum likelihood estimation (see previous section)." This should now be replaced by the uniform distribution as in L. 148: "The priors for the parameters are chosen from a wide uniform distribution"
I have no further comments.
Author Response
Answer to referee 1.
Note: L 299 still mentions: "Here we choose to use normally distributed priors inferred from the maximum likelihood estimation (see previous section)." This should now be replaced by the uniform distribution as in L. 148: "The priors for the parameters are chosen from a wide uniform distribution"
We completely agree. It was a mistake. We correct this sentence:
Here we choose to use uniform distributed priors (see previous section).